# Multiscale reconfiguration induced highly saturated poling in lead-free piezoceramics for giant energy conversion

Jinfeng Lin [1], Jin Qian[1], Guanglong Ge[1], Yuxuan Yang[2], Jiangfan Li[3], Xiao Wu[4], Guohui Li[1], Simin Wang[1], Yingchun Liu[5], Jialiang Zhang[3], Jiwei Zhai[1] ✉, Xiaoming Shi [6] ✉ & Haijun Wu [2] ✉

The development of high-performance lead-free $K_{0.5}Na_{0.5}NbO_3$-based piezoceramics for replacing commercial lead-containing counterparts is crucial for achieving environmentally sustainable society. Although the proposed new phase boundaries (NPB) can effectively improve the piezoelectricity of KNN-based ceramics, the difficulty of achieving saturated poling and the underlying multiscale structures resolution of their complex microstructures are urgent issues. Here, we employ a medium entropy strategy to design NPB and utilize texture engineering to induce crystal orientation. The developed $K_{0.5}Na_{0.5}NbO_3$-based ceramics enjoys both prominent piezoelectric performance and satisfactory Curie temperature, thus exhibiting an ultrahigh energy harvesting performance as well as excellent transducer performance, which is highly competitive in both lead-free and lead-based piezoceramics. Comprehensive structural analysis have ascertained that the field-induced efficient multiscale polarization configurations irreversible transitions greatly encourages high saturated poling. This study demonstrates a strategy for designing high-performance piezoceramics and establishes a close correlation between the piezoelectricty and the underlying multiscale structures.

Developing an efficient energy conversion material (e.g., electro-mechanical, acoustic-electrical) is a long-term pursuit for many cutting-edge applications, such as medical diagnosis and microelectromechanical systems (MEMS)[1,2]. Benefiting the unique piezoelectric effects, piezoceramics are widely used in transducers and energy harvesters in the aforementioned fields[3–5]. Unfortunately, in the current global market for piezoelectric devices, the lead-containing piezoelectric material segment holds a major share, especially the Pb(Zr, Ti)O₃ (PZT) series. Considering the hazards of lead, the compound annual growth rate of lead-free piezoelectric devices from 2019 to 2024 is expected to be 20.8%, but seldom sufficient[1,6]. The root cause

of their large gap in the market should be attributable to the inferior performance of lead-free piezoelectric materials. Under the joint blessing of Curie temperature and piezoelectricity, sodium potassium niobate ($K_{0.5}Na_{0.5}NbO_3$, KNN) based piezoceramics have become the most favored alternative component of lead ones in several well-known lead-free materials. Hence, substantial efforts have been undertaken to synergistically boost the comprehensive electromechanical characteristics of KNN-based piezoceramics to fulfill the pressing demands for lead-free electronic devices[6–8].

It is of utmost importance to gain a deeper insight into the structure-property relationship for the design of high-performance

[1]School of Materials Science and Engineering, Tongji University, Shanghai, China. [2]State Key Laboratory for Mechanical Behavior of Materials, Xi'an Jiaotong University, Xi'an, P. R. China. [3]School of Physics, State Key Laboratory of Crystal Materials, Shandong University, Jinan, China. [4]Key Laboratory of Eco-materials Advanced Technology, College of Materials Science and Engineering, Fuzhou University, Fuzhou, China. [5]Functional Materials and Acoustooptic Instruments Institute, Harbin Institute of Technology, Harbin, China. [6]Department of Physics, University of Science and Technology Beijing, Beijing, China. ✉e-mail: apzhai@tongji.edu.cn; shiming_870@163.com; wuhaijunnavy@xjtu.edu.cn

materials. Coexisting heterogeneous states of comparable energy within ferroic materials typically exhibit extraordinary responses to external field stimuli, such as giant electrocaloric effect and magnetostriction, which can be exploited in the design of advanced ferro/piezoelectrics[2,9]. For instance, the rhombohedral and tetragonal (R–T) phase coexisting at the morphotropic phase boundary (MPB) of PZT and Ba(Ti$_{0.8}$Zr$_{0.2}$)O$_3$-(Ba$_{0.7}$Ca$_{0.3}$)TiO$_3$ (BCTZ) ceramics brings up excellent piezoelectricity[10,11]. Although the origin of significant piezoelectric activity at the phase boundary is still controversial, almost all studies indicate the key to its existence due to the fact that the evolution of the internal macro/microstructure is sensitive to external electric field stimuli at the phase boundaries[8,12]. Pure KNN is known to have an orthorhombic perovskite-type structure with moderate piezoelectricity ($d_{33}$ ~ 120 pC/N) at room temperature. Actually, with cooling down from Curie temperature ($T_c$ ~ 420 °C), it also undergoes the following polymorphisms: low-temperature rhombohedral phase (R) and high-temperature tetragonal phase (T)[13].

Incipiently, inspired by MPB phase boundaries in PZT ceramics, the researchers mainly shifted $T_{O-T}$ or $T_{R-O}$ to near room temperature for constructing the O–T or R–O polymorphic phase boundary (PPB) via chemical modification, which significantly improved $d_{33}$ values of KNN-based ceramics to 400 pC/N[6–8,12]. Subsequently, both $T_{O-T}$ and $T_{R-O}$ were further tactfully moved to room temperature, resulting in the emergence of new phase boundaries (NPB) where R–O–T or R–T phases coexist and pushing the $d_{33}$ value of non-textured KNN-based ceramics to a new all-time high (~650 pC/N)[12,14]. Coincidentally, entropy increase strategies with multiple elements occupying equivalent lattice positions have also recently provided crucial clues for the component design of KNN-based ceramic with multiphase coexistence at the microscale[9,15,16]. However, the construction of phase boundaries also brings some new issues. On the one hand, achieving higher piezoelectricity inevitably requires sacrificing the Curie temperature. Furthermore, the presence of short-range polar nanoregions (PNRs) within the NPB is not entirely favorable to the piezoelectricity of KNN-based ceramics. Although PNRs facilitate domain switching, it is difficult to achieve saturated poling due to the large energy difference between PNRs and the ferroelectric matrix as well as the high content of non collinear PNRs, thus, a satisfactory mechanical coupling factor ($k_p$ < 0.6) cannot be achieved[8]. On the contrary, crystal anisotropy of piezoceramics based on texture engineering can not only improve the $d_{33}$ lead-free piezoceramics without sacrificing the Curie temperature but also greatly improve their mechanical coupling properties, which precisely compensates for the shortcomings brought by phase boundary engineering[17,18].

Inspired by the aforementioned considerations, we focus on the preparation of high-performance lead-free KNN-based piezoceramics by virtue of multiscale reconfiguration and try to reveal the structural/physical origin of high piezoelectric response. In this work (Fig. 1), the recently emerging entropy-dominated phase transition strategy is proposed to design NPB with local polymorphic distortion. Several classical phase boundaries moving additives (Ca$^{2+}$, Bi$^{3+}$, Hf$^{4+}$, Zr$^{4+}$, Ti$^{4+}$, Sb$^{5+}$) are introduced into the A/B site of K$_{0.5}$Na$_{0.5}$NbO$_3$ lattices to enhance the entropy value of the system. Meanwhile, the templated grain growth (TGG) technology was applied to induce the crystal orientation. As a result, the developed mesentropic textured KNN-based ceramics enjoys both prominent piezoelectric coefficient and mechanical coupling factor ($d_{33}$ ~ 680 ± 35 pC/N, $k_p$ ~ 72.5%), as well as high $T_c$ ~ 260 °C, thus exhibiting great potential in energy conversion. For example, the prepared vibration energy harvester can yield ultrahigh energy harvesting performance ($W_{out}$ ~ 4.00 mW, $P_D$ ~ 57.90 μW/mm³), and the prepared ultrasonic transducer achieved a broad −6 dB bandwidth (≈55.6%) and low insertion loss (IL ≈ −37.0 dB) with a center frequency of 2.15 MHz ($f_c$), outperforming most lead-free piezoceramics and even some commercial lead-based piezoceramics.

## Results

### Multiscale reconfiguration

Starting from the orthorhombic perovskite-type K$_{0.5}$Na$_{0.5}$NbO$_3$ matrix (Fig. S1a), the mesentropic piezoelectrics were designed by introducing Sb, Zr, Hf and Ti elements into the B-sites, and Ca and Bi into the A-sites, respectively, with the nominal composition of (K$_{0.505}$Na$_{0.5(0.99-x\%)}$Ca$_{0.01}$Bi$_{0.5-x\%}$)(Nb$_{0.965(0.99-x\%)}$Sb$_{0.035(0.99-x\%)}$Zr$_{0.01}$Hf$_{0.98-x\%}$Ti$_{0.02-x\%}$)O$_3$ ($x = 0 - 7$, abbreviated as $x$BHT). The atomic configuration entropy $S_{config}$ is defined as[15,16]

$$-R\left(\left(\sum_{i=1}^{N} x_i \ln x_i\right)_{cation-site} + \left(\sum_{j=1}^{M} x_j \ln x_j\right)_{anion-site}\right) \quad (1)$$

where $R$, $N/M$ and $x_i/x_j$ denote the ideal gas constant, atomic species and contents at the cation/anion sites, respectively. When 1 R < $S_{config}$ < 1.5 R, it is defined as medium entropy, while $S_{config}$ ≥ 1.5 R is defined as high entropy, otherwise it is low entropy. As displayed in Fig. S1b, the entropy of pure KNN ceramics increases from 0.7 R to nearly 1 R when the Ca, Zr and Sb elements are introduced, and then successfully enter the medium-entropy region with the help of Bi, Hf, and Ti elements, such as 1.2 R at $x = 4$, and even increases to 1.37 R when $x = 7$.

As expected, the increase of entropy induces the occurrence of phase transitions. Due to the low entropy value, X-ray diffraction patterns (XRD, Fig. 2a$_1$) demonstrate that the random R-0BHT ceramics doped only with Zr and Ca exhibit the same orthorhombic perovskite-type structure as pure KNN. When Bi$^{3+}$/Hf$^{4+}$/Ti$^{4+}$ were further introduced, due to the increase in entropy, the splitting diffraction peak around 45° began to aberrance, i.e., the (200) peak gradually strengthening while (002) one gradually weakens, resulting in the coexistence of R–O–T multiphase (NPB) near the R-4BHT ceramic[9,14,19]. As the entropy gradually approaches the higher entropy region (e.g., $S_{config}$ = 1.37, R-7BHT, Fig. S1b), the splitting characteristic of the diffraction peak disappears, indicating a highly symmetric phase structure (Fig. 2a$_1$). The local structural evolution can also be reflected by the Raman spectroscopy[20,21]. These Raman scattering peaks are mainly caused by different vibration modes of NbO$_6$ octahedra, including stretching modes ($v_1$ and $v_2$) and bending modes ($v_5$). For example, both $v_1$ and $v_5$ degenerate symmetric vibrational modes are very sensitive to the phase structure of KNN-based piezoceramics. Two important phenomena can be observed with increasing entropy: 1. The relative intensity of the $v_1 + v_5$ scattering peaks $\left(I_{v_1+v_5}\right)$ decreases gradually and the half-peak width increases, indicating that the increase in entropy leads to an increase in local disorder, i.e., an increase in local relaxor feature (Fig. S2a). 2. The ratios of $I_{v_2}$:$I_{v_1}$ as well as $I_{v_6}$:$I_{v_5}$ turn around at $x > 3$, indicating the occurrence of phase transitions, i.e., NPB (Fig. S2c). To dynamically demonstrate the formation of NPB, the curves of temperature-dependent dielectric constants ($\varepsilon_r$–T) are presented in Fig. S3a. As Bi$^{3+}$/Hf$^{4+}$/Ti$^{4+}$ increases, the two separated dielectric anomaly peaks in low entropy R-0BHT (−60 °C for $T_{R-O}$ and 145 °C for $T_{O-T}$) gradually approach and merge together near room temperature to form $T_{R-O-T}$ in R-4BHT. At the same time, the originally prominent dielectric peaks (e.g., $T_{O-T}$, and $T_C$) also gradually diffuse, and the Curie temperature moves toward room temperature, while $\varepsilon_r$ at room temperature first increases and then decreases, indicating that the long-range ordered domains are destroyed. When approaching the higher entropy region, the $T_{R-O-T}$ of R-7BHT disappears, leaving almost only the Curie peak near room temperature, suggesting that the phase structure deviates from NPB and approaches the paraelectric phase, namely, pseudocubic structure. Hence, the phase stability effect of entropy increase plays an essential role in the phase evolution induced in $x$BHT ceramics by Bi$^{3+}$/Hf$^{4+}$/Ti$^{4+}$ doping. Based on these analyzes of XRD patterns, $\varepsilon_r$–T curves

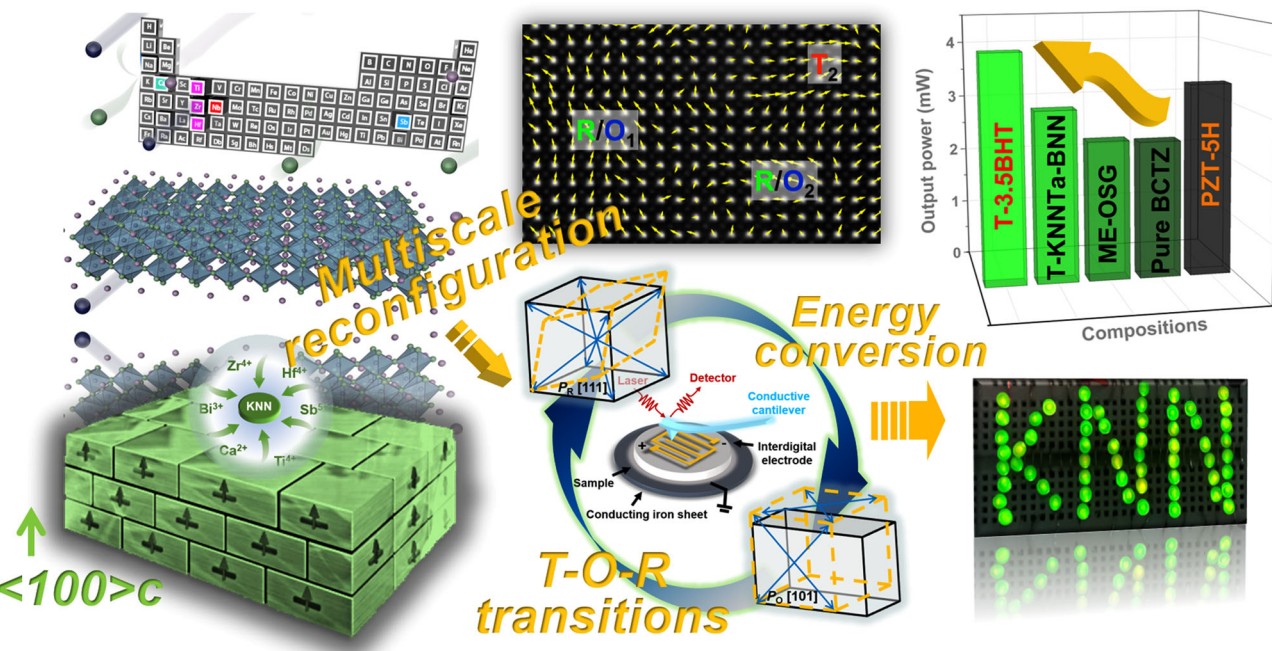

**Fig. 1 | Schematic diagram.** The multiscale reconfiguration strategy for high-performance KNN-based piezoceramics.

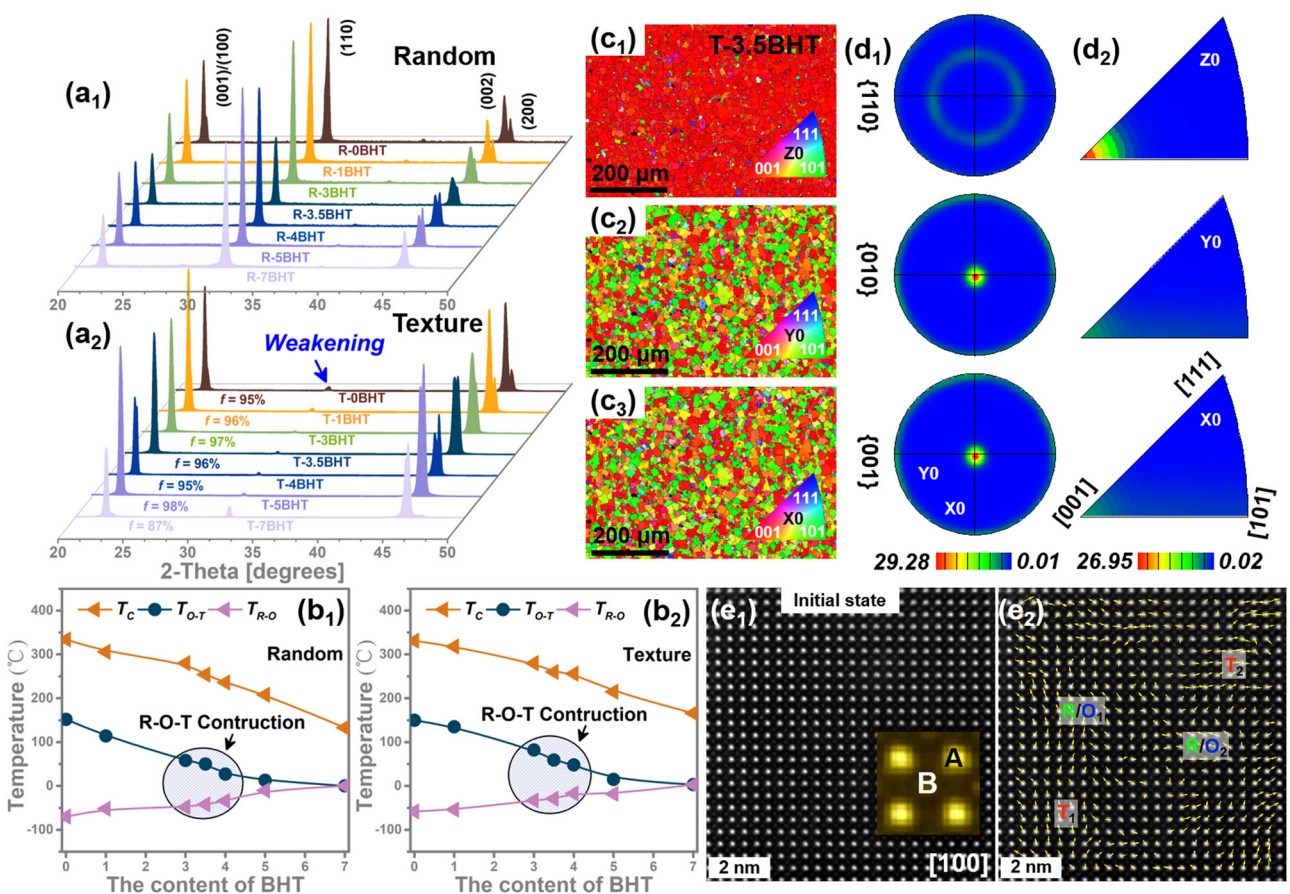

**Fig. 2 | The textured mesentropic *x*BHT ceramics.** XRD patterns of the random (**a₁**) and textured (**a₂**) *x*BHT ceramics. Composition-temperature phase diagrams of the random (**b₁**) and textured (**b₂**) *x*BHT ceramics. Inverse pole figure maps of *<00l>* c textured T-3.5BHT ceramics along (**c₁**) Z, (**c₂**) Y and (**c₃**) X-axis. **d₁** Pole and (**d₂**) inverse pole figure set of the T-3.5BHT ceramics. **e₁** Atomic-resolution scanning transmission electron microscopy high-angle annular dark-field (STEM-HAADF) polarization vector image along [100] zone axis for the unpoled T-3.5BHT, and (**e₂**) corresponding overlaid colorized displacement (polarization) vector map.

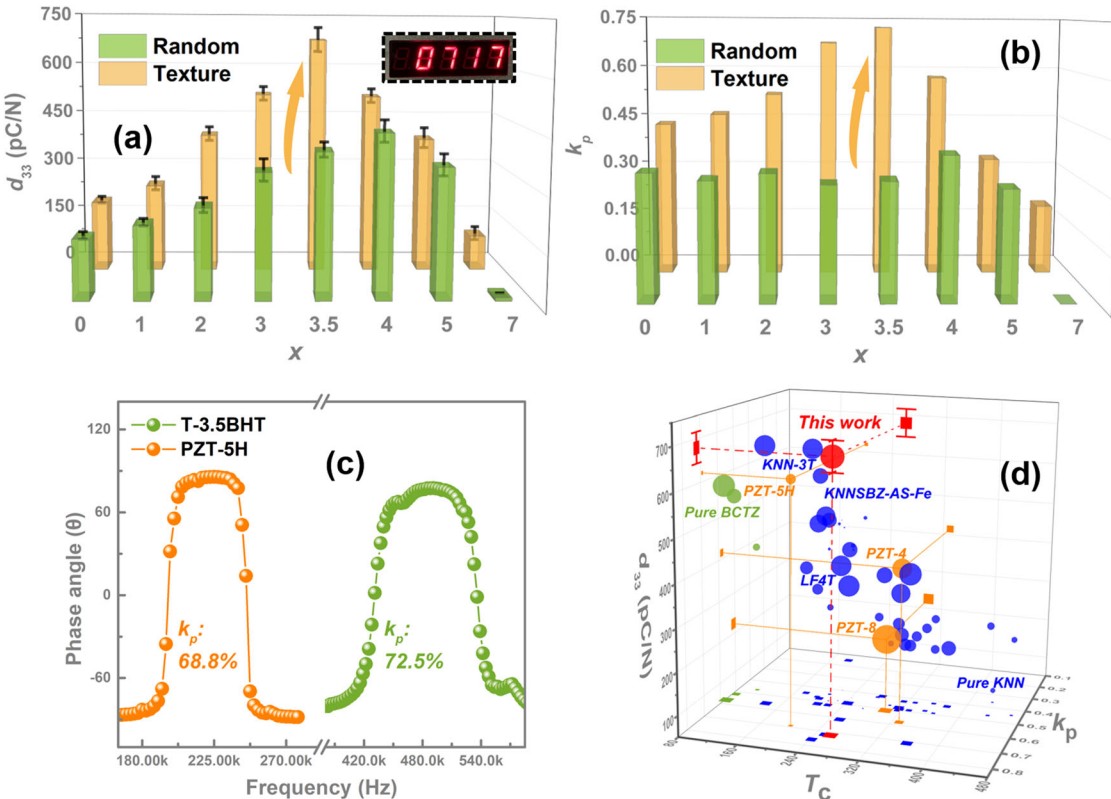

**Fig. 3 | Piezoelectric properties. a** The piezoelectric coefficient $d_{33}$ of the random and textured $x$BHT ceramics. **b** The planar electromechanical coupling factor $k_p$ of the random and textured $x$BHT ceramics. **c** Comparison of $k_p$ between the T-3.5BHT and the PZT-5H ceramics. **d** The comprehensive performance comparison of the T-3.5BHT ceramics with other lead-free/lead-based piezoelectric ceramics, including the $d_{33}$, $k_p$ and $T_c$.

and Raman spectra, the corresponding phase diagrams of the R-$x$BHT ceramics was established, as depicted in Fig. 2b$_1$.

Noticeably, previous experiments have shown that the increase in the content of Bi, Hf and Ti elements not only increases the sintering temperature but severely suppresses the temperature range of grain growth, which greatly hinders the texture progress in higher entropy components (e.g., T-3.5BHT, Fig. S4c)[9]. After extensive experiments, it was found that lowering the calcination temperature of the ceramic powders favored higher activity, and thus higher entropy $x$BHT textured ceramics were successfully prepared. The textured $x$BHT ceramics are not only homogeneous with no deviation in content (Fig. S5–S7) but also exhibit uniformly larger sized and brick-wall grains aligned along <00*l*>c (Fig. S4e, f). Hence, the intensity of <00*l*>c peaks in the T-$x$BHT ceramics increases comprehensively, while the intensity of <110 > c peaks significantly weakens, resulting in a calculated texture degree of over 95% (Fig. 2a$_2$). Subsequently, the electron back scatter diffraction (EBSD) technology was used to further evaluate the quality of texture (Fig. 2c, d). From the inverse pole figure (IPF) maps and inverse pole/pole figure (IPF/PF) set, it can be seen that the orientation distribution of polycrystals for the T-3.5BHT deviates from random distribution and shows some regularity. For example, almost all grains' <00*l*>c directions are parallel to the Z-axis of the sample (perpendicular to the casting direction) instead of X and Y-axis (Fig. 2c), and the $M_{max}$ values in the both IPF/PF set are also greater than 25 (Fig. 2d), indicating a strong <00 *l* > c texture degree for the T-3.5BHT[18,22]. According to the comprehensive analysis of $\varepsilon_r$–T curves, room temperature XRD patterns, and Raman spectroscopy (Figs. 2a$_2$, S2b, d and S3b), the phase structure evolution of the T-$x$BHT ceramics is similar to that of R-$x$BHT with the increase of $x$ (Fig. 2c). However, due to the addition of extra NN and differences in grain size, the NPB composition of the T-$x$BHT ceramics is advanced to T-3.5BHT. Because not only can we see from XRD that the ratio of (200) peak to

(200) of T-3.5BHT is similar to that of R-4BHT (Fig. 2a), but also $T_{R-O}$ at low temperature in the $\varepsilon_r$–T curves also disappeared in advance in T-3.5BHT, while there is still evident in R-3.5BHT (Fig. S3). We also observed the dynamic changes in the phase structures of T-$x$BHT with respect to composition and temperature via in situ XRD and Raman spectroscopy (Fig. S8). As a result of the phase transitions, the peaks of Raman spectroscopy and XRD show significant abrupt changes with temperature, corresponding to $T_{R-O}$, $T_{O-T}$, and $T_c$. In agreement with the $\varepsilon_r$–T curves, it can be seen that the $T_{R-O}$ and $T_{O-T}$ in T-0BHT are merged at room temperature to ~70 °C to form NPB at T-3.5BHT, and the $T_c$ is also reduced from ~350 °C to ~260 °C (Figs. S8b, e, f). The formation of NPB tends to induce local polymorphic distortion[9,14,16]. Fig. 2e$_1$ gives one representative STEM high-angle annular dark-field (HAADF) image of the unpoled T-3.5BHT along the [100] zone axis. The polarization vectors are determined by the atom displacement from B-site cations (weaker intensity contrast) to the center of the four nearest neighboring A-site cations (stronger intensity contrast) based on 2D Gaussian peak fitting. Fig. 2e$_2$ is atomically resolved STEM HAADF images along [100], superimposed with a map of atom polarization vectors (indicated by yellow arrows). As expected, R, O, and T phases correlated polarization vectors can be clearly resolved. The coexistence of such multiphase correlated polarization vectors with nearly isotropic free energies significantly reduces the polarization anisotropy and predicts an efficient polarization response in response to an external stimulus (i.e., electric fields).

## Structure-dependent ferro/piezoelectric properties

We measured P-E hysteresis loops and strain-electric field (S-E) curves for the $x$BHT ceramics. As entropy increases, the random fields brought by doped ions gradually break the long-range order of ferroelectric macrodomains[6,20], which not only improves the dielectric response (Fig. S3) but also promotes the flattening of free energy,

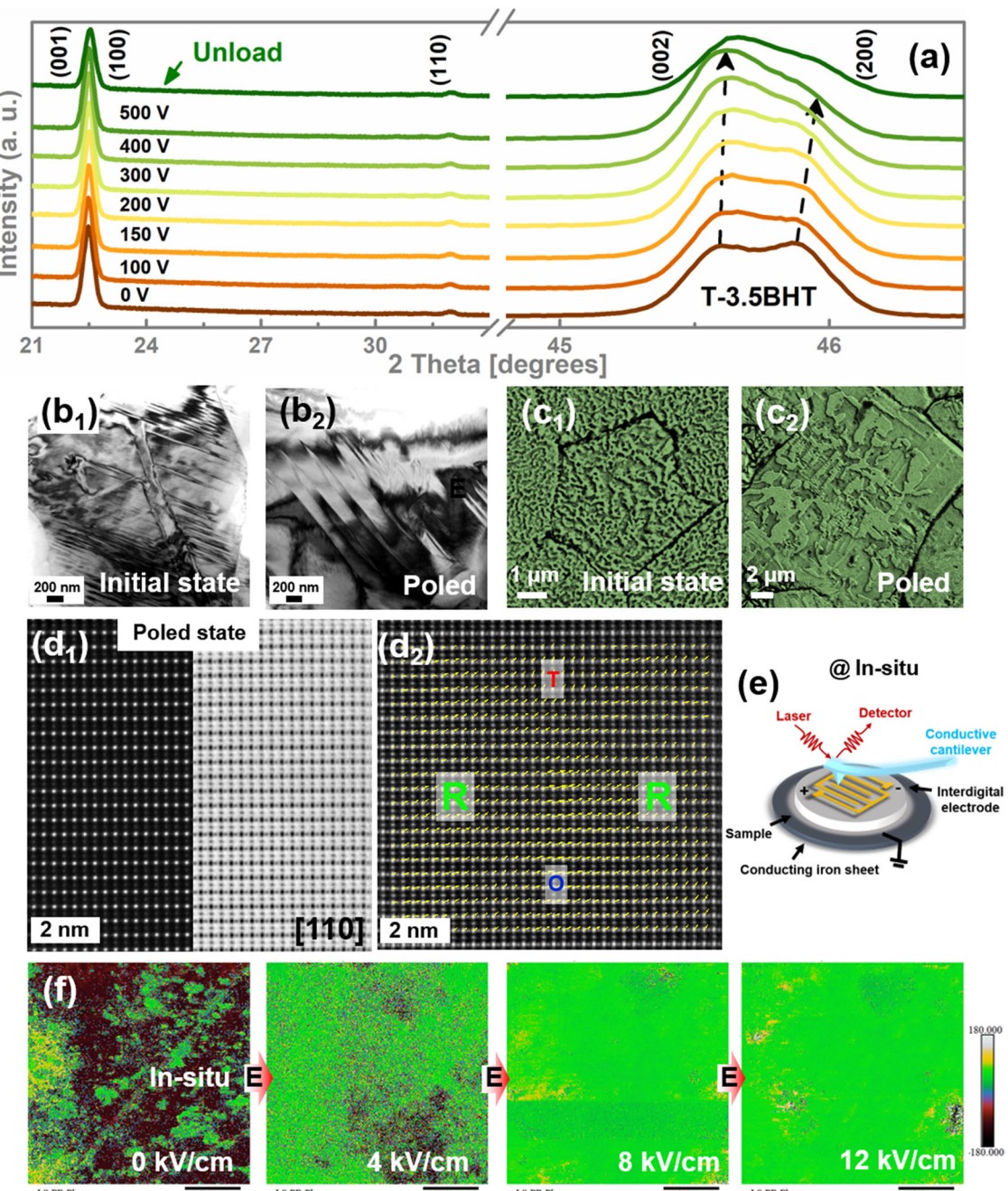

**Fig. 4 | Electric field-induced multiscale polarization configurations transformation. a** In situ electric field dependent XRD for T-3.5BHT ceramics. **b** Transmission electron microscopy (TEM) bright-field images and (**c**) SEM images of acid-etched domain for unpoled and poled T-3.5BHT ceramics. **d₁** STEM-HAADF and ABF images along [110] zone axis for the poled T-3.5BHT, and **d₂** corresponding overlaid colorized displacement (polarization) vector map. **e** Sketch map of the interdigital electrode for testing the in situ piezoresponse force microscopy (PFM) via applied electric field. **f** In situ electric field dependent optimized vertical PFM for T-3.5BHT ceramics.

resulting in a continuous decrease in the coercive field ($E_c$) of the $x$BHT ceramics (Fig. S9a, b). The enhanced dielectric response (Fig. S3) and domain switching capability greatly improve the ferro/piezoelectricity of the $x$BHT ceramics, especially at the NPB (e.g., R-4BHT and T-3.5BHT). However, the high crystal symmetry and the disappearance of ferroelectric macrodomains increase the relaxor characteristics of higher entropy components (i.e., $x \geq 5$), leading to a deterioration of ferro-/piezoelectricity. As a result, the $d_{33}$ values and corresponding electric field-induced positive/negative strain of the $x$BHT ceramics first increase and then decrease with the increase of $x$, and the optimum is reached at NPB (Fig. 3a and S9). Notably, compared to random ceramics, the ferroelectricity and electrostrain of textured ceramics

are significantly improved (Figs. S9a, b and S10a, b). The positive and negative electrostrain of T-3.5BHT can reach nearly ~ 0.2% and ~-0.19% under 30 kV/cm, respectively, while that of R-4BHT are only ~ 0.15% and ~-0.07%, indicating high piezoelectric activity of T-3.5BHT (Fig. S10b). Hence, from Fig. 3a, it can be seen that on the basis of existing phase boundary, the $d_{33}$ of T-3.5BHT has been further improved from ~ 450 ± 30 pC/N to an ultra-high value of ~ 680 ± 35 pC/N. Meanwhile, the planar mechanical coupling factor ($k_p$) was also successfully increased from ~ 40.0% to ~72.5% (Fig. 3b, c), well in line with our original intention. More importantly, the improvement of piezoelectric performance did not sacrifice the $T_c$ of T-3.5BHT (~260 °C), allowing its $d_{33}$ to remain above ~600 pC/N before 200 °C,

which is beneficial for practical applications (Fig. S11). The phase angle usually reflects the polarization state of piezoceramics. After texturing, the phase angle has been significantly increased, with up to 80 in the components of T-$x$BHT ceramics with $x < 5$, indicating a highly saturated polarization state (Fig. S12). This is because the crystal orientation makes the arrangement of polarity vectors more effective under the applied electric field, thereby improving both $d_{33}$ and $k_{\rm p}$ simultaneously[23]. However, in the higher entropy components (i.e., $x \geq 5$), due to the increase in crystal symmetry and relaxor characteristics, saturated polarization cannot be achieved in either the texture or random ceramics. By comparing $d_{33}$, $k_{\rm p}$, and $T_{\rm c}$, it can be seen that the comprehensive performance of the prepared T-3.5BHT ceramic is highly competitive in both lead-free piezoelectric ceramics and commercial lead-based ceramics (Fig. 3d)[9,17–19,21,23–32]. More importantly, due to the elastoelectric composite effect and the discrepancy in electrical properties between the NN templates and KNN matrix[33], it can be found that the piezoelectricity of the T-$x$BHT ceramics are effectively increased without increasing its dielectric constant (Fig. S3), which is favorable for the high piezoelectric voltage coefficient $g_{33}$ (a key factor for piezoelectric receiving transducers) and the high piezoelectric activity quality factors ($d_{33} \times g_{33}$, FOMs for piezoelectric energy harvesters)[3].

For better elucidate the deeper factors of piezoelectric property enhancement, including $d_{33}$ and $k_{\rm p}$, we further tested the $P$-$E$ loops and unipolar $S$-$E$ curves under different electric fields, as well the in situ bias voltage dielectric curves. Figure S10c, d display the unipolar $S$-$E$ curves under different electric fields and corresponding piezoelectric strain coefficient $d_{33}^{*}$ ($S$/$E$). Compared with the R-4BHT ceramics, the T-3.5BHT ceramics obtained a maximum value of $d_{33}^{*}$ in advance under a lower electric field (i.e., 914 pm/V at 10 kV/cm), which further indicates that the crystal orientation effectively promotes the alignment of polarity vectors, resulting in large lattice distortion under the low electric field. Furthermore, although the $\varepsilon_{\rm r}$ of textured ceramics is low, there is a significant hysteresis for the in situ bias voltage dielectric curves, especially in the positive electric field part (Fig. S10f), indicating that the electric field-induced phase/domain structure evolution in textured ceramics is more pronounced compared to their random counterparts, and thus intrinsic/extrinsic contribution of domain switching and phase transition under the action of the electric field is the key point of the high piezoelectric activity for the textured ceramics. Generally, Rayleigh analysis method is highly effective for analyzing the intrinsic or extrinsic contributions of piezoelectric and dielectric properties, which is obtained by calculations based on the $P$-$E$ loops of different electric fields below the coercive field (Fig. S13)[9,23]. The calculation results revealed that the construction of NPB greatly enhances the intrinsic contribution (lattice distortion) of the dielectric response for the $x$BHT ceramics, while the crystal orientation effectively increases its non-intrinsic contribution (Fig. S10e). Therefore, it is precisely due to the synergistic optimization effect between the free energy flattening brought by phase boundary construction and high-density domain walls generated by crystal orientation that the mesentropic T-3.5BHT achieves prominent piezoelectric comprehensive performance ($d_{33} \sim 680 \pm 35$ pC/N, $k_{\rm p} \sim 72.5\%$), and thus has potential for electromechanical applications.

## Decipher of electric field-induced multiscale local structure evolution

The piezoelectricity is closely related to the electric field-induced local structural phase transition. Figure 4a shows the detailed evolution of XRD patterns of T-3.5BHT when applying electric field. With the increase of the external electric field, the T-3.5BHT ceramics produces a large distortion, and the intensity of its (200) peak is rapidly exceeded by the (002) peak or even disappears, and tends to stabilize after 500 V (thickness ~0.3 mm), indicating a sequential phase transition process from T phase to O phase, and then to R phase. Thus, the construction of

NPB promotes the flattening of free energy and provides a convenient platform for the electric field-induced phase transition between R-O-T. Moreover, when the electric field is removed, its distortion only rebounds slightly. Compared to R-4BHT (Fig. S14), the phase transition of T-3.5BHT is more agile and complete with respect to the electric field, which is well consistent to its high saturated poling state. Although the significant macroscopic T-O-R sequential phase transition induced by electric field did not improve the intrinsic contribution of dielectric response in T-3.5BHT ceramics (Fig. S10 e), it is crucial in the intrinsic contribution of piezoelectric response. Besides the macroscopic ferroelectric phase, there are abundant ferroelectric domains related to the phase structure in the microscopic size of piezoelectric materials[6,34]. Apparently, with the increase of entropy, the large-sized block ferroelectric macrodomains associated with the orthorhombic phase of the 0BHT are initially destroyed into smaller sized watermark or stripe macrodomains. Eventually, a large number of weakly polarity polar nanoregions (PNRs) appear in the components close to the high-entropy region (i.e., $x = 7$, Fig. S15a–f). Hence, the domain switching becomes more pronounced under the applied -10 V tip bias at $x > 2$ due to the lower domain wall energy (Fig. S15d$_1$–f$_1$), which corresponds to the $E_{\rm c}$. It is precisely because a large number of smaller sized stripe macrodomains undergo irreversible switching under the applied electric field that the components at NPB achieve excellent piezoelectricity. However, although PNRs facilitates the domain switching, the presence of a large number of weakly polarity PNRs enhances the relaxor feature and deteriorates the polarizability, worsening the piezoelectricity instead (i.e., $x = 7$, Fig. S15c, f)[28,35–37].

Notably, further comparison reveals that the textured T-3.5BHT has thinner striped domains with higher domain wall density than R-4BHT (Fig. S15b, e). To reveal the microstructure of the domains, representative SEM images of acid-etched domain patterns and TEM images are shown in Fig. 4b, c, S16 and S17. Intuitively, high density of thin stripe macrodomains do exist in the large grains of the T-3.5BHT ceramics (Fig. 4b$_1$, c$_1$, S16c$_1$-c$_3$ and S17a), whereas larger sized polymorphic macrodomains (including watermark and hierarchical domains) are exhibited in the small grains of the R-4BHT ceramics (Fig. S16a$_1$–a$_3$). Thus, under the auspices of high domain wall density, high crystal orientation, as well as less grain boundary hindrance, the polarized growth of domains becomes more significant and the phase of the domain for the poled T-3.5BHT is more unified, indicating a more saturated polarization state (Fig. 4b$_2$, c$_2$, S16d$_1$–d$_3$, S17b and S18d). Note here that we also observe evidence of a transformation from the O/T-phase polarization vectors to R phase in the poled T-3.5BHT at the atomic level. This is because there is a microregion structure dominated by R-phase polarization vectors in the poled T-3.5BHT, with only trace amounts of O/T-phase polarization vectors are embedded in the R-phase matrix (Fig. 4d$_1$, d$_2$), which contrasts with the initial one that have a higher proportion of T-phase polarization vectors (Fig. 2e), thus further validating the electric-field-induced phase transition behavior of T-3.5BHT. To deeply track the growth of domains, interdigital electrodes were designed on the surface of the T-3.5BHT for in situ electric field PFM testing (Fig. 4e). On the one hand, it can be seen that the domains respond as sensitively as XRD at lower electric fields (Fig. 4a), e.g., the phase of the domains in T-3.5BHT is significantly pulled to 0 degrees (in the same direction as the electric field) before the applied electric field reaches $E_{\rm c}$ (Fig. 4f and S19). The difference is that the phase of the domains is almost saturated by 12 kV/cm, while the distortion of XRD diffraction peaks is delayed after 12 kV/cm, which is similar to previous experimental results[9]. The incomplete consistency between electric field induced irreversible domain switching and lattice distortion precisely corresponds to the two contributions of dielectric/piezoelectric responses (i.e., ex/intrinsic contributions)[9,23]. Thus, the much smaller domain-switching electric field of the T-3.5BHT than that of R-4BHT intuitively reveals that the large number of macroscopic domains irreversibly switched facilitated

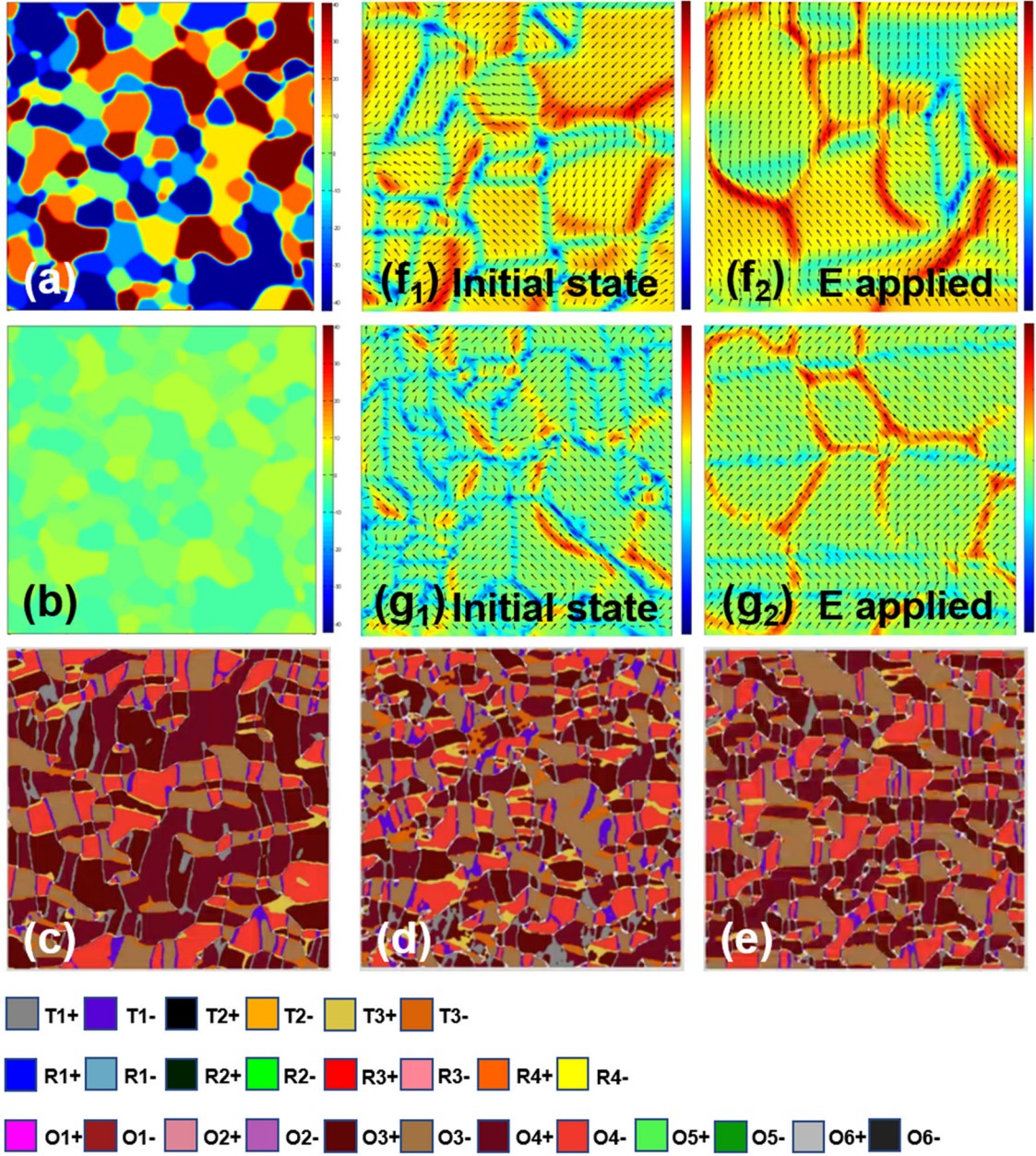

**Fig. 5 | Phase-field simulations.** Grain structures of random ceramics (**a**) and textured ceramics (**b**) with the same grain size, respectively. The domain structures for (**c**) pure KNN ceramics with single O phase, and (**d**) random ceramics and (**e**) texture ceramics with R–O–T three phase coexistence. The projection of polarization on the vertical (100) plan for (**f**) random ceramics and (**g**) textured ceramics with R-O-T three phase coexistence before and after appling eletric field (**e**).

by the high domain wall density, high crystal orientation, as well as less grain boundary hindrance effectively enhances the non-intrinsic contributions, thus greatly improving the saturated poling and piezoelectric response of the T-3.5BHT.

In order to enrich the relevant theories, we further elucidate the piezoelectric response through phase-field simulation. Several effects were considered in our model. Firstly, a ferroelectric polycrystal structure (KNN system) was generated, resulting in textured or random orientations at different grains. Figure 5a, b show the grain structures of random ceramics and texture ceramics with the same grain size but different orientation distributions, respectively. Secondly, considering the influence of multi-element doping on domain morphology involved in entropy modulation, a defect-related random electric field is also introduced[38,39]. In agreement with the experimental results, initially there are abundant large-sized domains of O phase (12 states) in pure KNN ceramics, and then are decomposed into high

density small-sized macroscopic domains with R, O, and T phases (8 R, 6 T, and 12 O states) coexist in both random ceramics and texture ceramics, which greatly increases the possible states of the spontaneous polarization ($P_s$) (Fig. 5c–e)[14]. In this case, the energy barrier of polarization rotation could be significantly reduced, benefiting the macrodomain switching and piezoelectric response. After the domain structure evolution is stable, an electric field up to 30 kV/cm was applied to investigate the polarization response along the Z direction (Fig. 5f, g). Due to the different ferroelectric anisotropy of different grains in random ceramics, the polarization direction exhibits strong non-uniformity. Therefore, after the electric field applied, there are still some domain structures that have not been completely switched. While the textured ceramics exhibit a more saturated polarization state than the random one. Hence, the simulation results further reveal that the multiscale reconfiguration based on phase boundary and crystal orientation plays a crucial role in electric field-induced domain

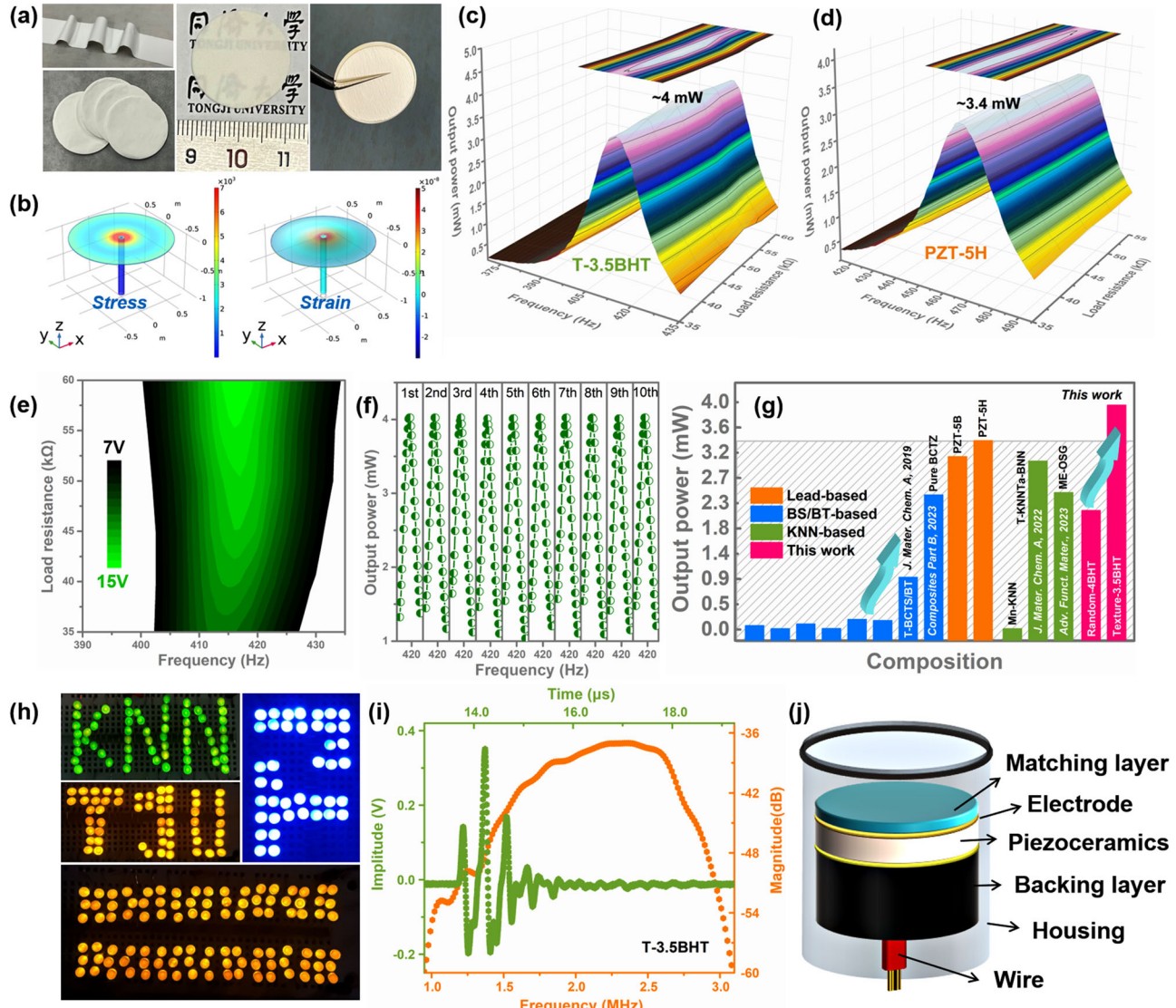

**Fig. 6 | Applications for energy conversion. a** Preparation process for the PCD piezoelectric energy harvester, including tape casting, slicing, sintering and silvering. **b** Simulation results of the stress/strain distribution of the PCD piezoelectric energy harvester (PCD PEH). The output power under different frequencies and different load resistances for the (**c**) T-3.5BHT and **d** PZT-5H ceramics. **e** The output voltage under different frequencies and different load resistances for the T-3.5BHT ceramics. **f** The output power with a load resistance of 42 kΩ measured 10 times. **g** The output power comparison of the T-3.5BHT PCD PEH with other lead-free/lead-based piezoelectric harvesters. **h** The real-time lighting photo of LEDs driven by the T-3.5BHT PCD PEH. **i** Pulse-echo waveform and frequency spectra of the T-3.5BHT ceramic transducer. **j** Structure schematic of transducer.

response and piezoelectric performance. Overall, the electric field-induced efficient sequential phase transition brings about an improvement in the intrinsic contribution to the piezoelectric response, whereas the electric field-induced efficient polarization switching of high-density domain walls leads to an increase in non-intrinsic contribution to the dielectric/piezoelectric response.

**Energy conversion of the T-3.5BHT ceramics**

To demonstrate the application feasibility of the designed high-performance mesentropic T-3.5BHT ceramics, piezoelectric energy harvesting and ultrasonic transducers are used as examples for electro-mechanical conversion. For piezoelectric energy harvesters (PEH), the generating output power ($P_{out}$) is directly proportional to $d_{33}^2/\varepsilon_r$, which is defined as the figure of merit (FOM)[40]. From the above clarifying, the $\varepsilon_r$ of the $x$BHT ceramics do not increase after texturing, while the $d_{33}$ has been significantly improved (especially for T-3.5BHT), which greatly benefits $P_{out}$. The well-sintered large-sized $x$BHT circular ceramics were manufactured into a piezoelectric circular diaphragm

(PCD) energy harvester to generate electricity, and the corresponding partial preparation process and schematic diagram are shown in Fig. 6a and S20a, b. From the cross-sectional SEM images, it can be seen that except for the silver electrodes on both sides, the thickness of the T-3.5BHT in the prepared energy harvester is ~0.22 mm (Fig. S20c). The simulation results of stress/strain distribution during the vibration process of the PCD energy harvester reflect the presence of high stress/strain in the central region of the sample, which is responsible for the main energy output (Fig. 6b). The $V_{out}$ and $P_{out}$ of the T-3.5BHT PEH under different frequencies/load resistances are shown in Fig. 6c, e. It can be seen that under the optimal load resistance and frequency conditions, the T-3.5BHT PEH can produce the maximum instantaneous $V_{out}$ and $P_{out}$ of ~13.37 V and ~4.00 mW, respectively, with good repeatability (Fig. 6f). More importantly, the calculated high power density $P_D$ ~ 57.90 μW/mm³ represents an advanced level in lead-free KNN based ceramics and even better than the commercial PZT-5H/PZT-5B and most other lead-free ones[9,18,27,41–53]. As a result, more than 100 light-emitting diodes (LEDs) can be easily lit

with its excellent electro-mechanical conversion performance, and the switches of these LEDs can also be sensitively controlled by vibration conversation (Supplementary Movie 1). As for the application of ultrasonic transducers, the T-3.5BHT with both high coupling factor and piezoelectric coefficient is favorable for the wide bandwidth and high sensitivity. Similarly, the well-sintered large-sized $x$BHT square ceramics were manufactured into an ultrasonic transducers, and the corresponding preparation process and schematic diagram are shown in Fig. 6j and S21a, b. Like other lead-free piezoelectric materials[54], due to the difficulty of fabricating large enough uniform dense T-3.5BHT ceramics, the piezoelectric properties are greatly discounted (e.g., $d_{33}$: 500–580 pC/N, $k_p$: 0.5–0.55). Even so, the T-3.5BHT-based transducer achieves a broad -6 dB bandwidth ($\approx 55.6\%$) and low insertion loss (IL $\approx$ -37.0 dB) with a center frequency of 2.15 MHz ($f_c$), which is comparable to that reported for lead-based single element phased array ultrasonic transducers (Fig. 6i)[55]. Thus, the designed high-performance mesentropic T-3.5BHT ceramics shows great potential for application in the field of electro-mechanical conversion.

In summary, giant piezoelectric properties with $d_{33}$ ~ 680 ± 35 pC/N and $k_p$ ~ 72.5% as well as high $T_c$ ~ 260 °C were obtained in the developed mesentropic T-3.5BHT ceramics. Benefiting from the formation of NPB and coarse grains with <00l> c orientation, atomic-scale polymorphic distortion and micrometer-scale high-density thin striped domains were discovered. A further comprehensive analysis of electric field dependent multiscale structure reveals that such microstructures dramatically reduce the free energy and efficiently promote the irreversible transformation of the multiscale polarization configurations, resulting in a high saturated poling. As a consequence, the T-3.5BHT ceramic shows prominent device ratings, e.g., ultrahigh energy harvesting performance as well as excellent transducer performance. This work addresses the difficulty of saturated poling of NPB-type KNN-based ceramics and facilitates the development of lead-free piezoceramic materials in energy conversion applications. Furthermore, the multiscale structural measurements carried out in this study also provide a feasible route for analyzing complex phase boundaries in other similar materials.

## Methods
The specific experimental details are provided in supporting information.

## Data availability
All data supporting this study and its findings are available within the article and its Supplementary Information. Any data deemed relevant are available from the corresponding author upon request. Source data are provided with this paper.

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

## Acknowledgements

This work was supported by the National Natural Science Foundation of China under Grant Nos. 52032007, 52172128 and 51772211, as well as the National Key R&D Program of China (2021YFB3201100).

## Author contributions

J.W.Z. and H.J.W. supervised this work. J.F.L. conceived the original idea and designed all aspects of the experiments. G.L.G. and G.H.L. performed the ex/in situ XRD experiments. J.Q., G.L.G. and S.M.W. performed the ex/in situ PFM characterizations. J.F.L. and J.L.Z. performed the acid-etched domain experiments. H.J.W. and Y.X.Y. performed the TEM and STEM experiment and quantitative analysis. X.M.S. performed phase-field simulations. Y.C.L. performed ultrasonic transducers experiments. J.F.L., X.W., H.J.W. and J.W.Z. revised the manuscript. All authors discussed the results. Additionally, we would like to thank Mrs. Yang Zhang from Instrument Analysis Center of Xi'an Jiaotong University for the assistance with electron microscopy characterization.

## Competing interests

The authors declare no competing interests.
