## [Peer Review File · Nature Communications]

Multiscale Reconfiguration Induced Highly Saturated Poling in Lead-Free Piezoceramics for Giant Energy ConversionREVIEWER COMMENTS

Reviewer #1 (Remarks to the Author):

This manuscript introduces a innovative approach that employs a medium entropy strategy and texture engineering to enhance the piezoelectric and mechanical coupling properties of lead-free KNN-based piezoceramics, achieving remarkable energy harvesting performance. The research establishes a strong correlation between piezoelectricity and multiscale structures, providing a promising design strategy for high-performance piezoceramics. These findings are intriguing, offering a new design principle of composition and texture engineering in lead-free piezoceramic materials and the resulting energy conversion devices.

However, I have several comments and questions:

In Figure 2, it is recommended to spell out the abbreviation EBSD when it first appears.

Additionally, please include captions for figures b1, b2, and b3, and provide more detailed discussions regarding the EBSD images and pole figures.

Consider including the data point for the 3.5BHT samples in the composition-temperature phase diagrams since these textured 3.5BHT ceramics exhibit the highest piezoelectric properties.

Could the authors specify the geometric dimensions of the samples used to test the planar electromechanical coupling factor? Also, it would be helpful if the authors could indicate the resonance and antiresonance frequencies for all samples in Fig. S10.

Regarding the comparison of polarization and strain loops, in-situ XRD patterns, SEM images, and PFM images of R-4BHT and T-3.5BHT ceramics, did the authors consider the potential impact of composition on the results?

How did the authors mitigate the influence of acid etching and the TEM sample preparation process on the domain structure before and after poling?

The temperature stability of the piezoelectric performance of the T-3.5BHT sample is not mentioned. Could the authors provide insights into this aspect?

Given that the transducer performance is closely tied to the fabrication process and the selection of matching and backing layers, could the authors elaborate on how they considered the selection of materials for the matching layer and backing layer? Additionally, what advantages does the transducer performance obtained in this manuscript have compared to that made by other lead-free piezoelectric materials?

Additional minor comments:

The image in Figure S6a appears to be blurry; kindly consider increasing the resolution of the picture for better clarity.

Regarding the statement, "When $1 < S_{\text{config}} < 1.5R$, it is defined as medium entropy," it is suggested to modify it to "When $1R < S_{\text{config}} < 1.5R, \dots$ " for clarity.

In Figure 4h2, please verify whether the annotation for E is redundant.

Reviewer #2 (Remarks to the Author):

Comments to the Author:

In this manuscript, the author comprehensively studied the effects of entropy and crystal orientation on the electrical properties of KNN-based ceramics through ingenious experimental design. Consequently, giant piezoelectric properties with $d_{33} \sim 680 \pm 35$ pC/N and $k_p \sim 72.5\%$, as well as high $T_c \sim 250$ °C, were obtained in the newly developed mesentropic T-3.5BHT ceramics. The results are very exciting, and I would recommend for

publications in NC after minor revisions.

1. With so many additives in the formula, what is the role of each component?
2. The dielectric constant of the textured ceramics is not higher than that of random ceramic components, and even lower. Can you further explain?
3. The schematic in Figure 1 shows that the grains of the textured ceramics grow along the vertical casting direction, can such information be seen in the SEM?
4. TEM is used to observe the evolution of the domain structure before and after polarization. Please describe the details. How do you ensure that the polarization state is not damaged during the sample preparation process and the characterization stage (e.g., by the electron beam)?
5. The stability of the piezoelectric properties over time is crucial. Can you do a cycling test for the piezoelectric properties?

Reviewer #3 (Remarks to the Author):

The development of high-performance lead-free KNN-based piezoceramics is very important for achieving environmentally sustainable society. This work employed a medium entropy strategy to design NPB and utilize texture engineering to induce crystal orientation. The newly developed KNN-based ceramics enjoys both prominent piezoelectricity and mechanical coupling factor ($d_{33} \sim 680 \pm 35$ pC/N, $k_p \sim 72.5\%$), thus exhibiting an ultrahigh energy harvesting performance ($W_{out} \sim 4.00$ mW, $PD \sim 57.90$ μ W/mm³) as well as excellent transducer performance. This work is interesting and the results are very good. The author also gives a deep insight explanation for the observed results. However, a mandatory revision is required. Here are some comments.

1. One of the main topics of this work is employing a medium entropy strategy to design NPB. It is still unclear what is the advantage of this strategy when comparing other methods of designing NPB? Because the d_{33} is even much lower than some reported KNN based materials with composition mediated NPB with d_{33} over 550 pC/N. Furthermore, what is the relationship between it and the following texture process.
2. The order of some figures are confusing. (1) The order of the figures in Figure 2 should be rearranged. According to the reading habit, the figure 2a is better on the left of Figure 2. (2) The Figures 4a and b for STEM appear before Figure 3, which makes the readers very confusing and eyes jump in very side of the manuscript during reading. The figures are suggested to rearranged to make the reader read easier and more comfortable.
3. Figure 4c and d are not described in the main text.
4. The chemical form may be wrong. If the x is larger than 1, the amount of some elements are minus, please check them carefully. Furthermore, from the chemical form, the change of the elements is very small, it may very hard to precisely to weight them. As we know, the phase boundary of KNN based material are very sensitive to the composition, how can you keep the consistence of the performance.
5. In page 10, the author said "Both the positive/negative electrostrain of T-3.5BHT reached nearly 0.2% under 30 kV/cm, while that of R-4BHT were only $\sim 0.15\%$ and $\sim -0.7\%$, respectively, indicating extremely high piezoelectric activity of T-3.5BHT". From Fig. S9b, no negative strain with value over -0.2% is observed, how can the large value -0.7% can be obtained? Is it -0.07% ? Please double check the data. Furthermore, the value of negative strain should also be given in this sentence for the composition T-3.5BHT.
6. The caption of the figures in Supporting information should be more detailedly described, similar to those of the figures in the main manuscript. Otherwise the reader will be very hard

to tell out of the figures, which will affect the wide spread of this work.

7. How can the texture engineering improve the k_p ? Please give more discussions.

8. There are some writing errors in the manuscript, which should be carefully checked. Here are some examples:

(1) In line 2 Page 17, "...while the d_{33} has been significantly improves (especially for T-3.5BHT) ...", the word "improves" should be "improved".

(2) In Page 12, "... which is well consistent with to its high saturation polarization state and high piezoelectricity..." the word "with" should be deleted.

Referee #1:

General comment:

“This manuscript introduces an innovative approach that employs a medium entropy strategy and texture engineering to enhance the piezoelectric and mechanical coupling properties of lead-free KNN-based piezoceramics, achieving remarkable energy harvesting performance. The research establishes a strong correlation between piezoelectricity and multiscale structures, providing a promising design strategy for high-performance piezoceramics. These findings are intriguing, offering a new design principle of composition and texture engineering in lead-free piezoceramic materials and the resulting energy conversion devices.

However, I have several comments and questions:”

General response:

We appreciate your positive and constructive comments and suggestions. We have studied the comments and tried our best to revise the manuscript according to your carefully and the corrections are listed below point by point:

All changes in the revised manuscript are highlighted in red.

The followings are the responses for the comments by the reviewers for the manuscript (NCOMMS-23-62921):

Detailed comment 1: “In Figure 2, it is recommended to spell out the abbreviation EBSD when it first appears. Additionally, please include captions for figures b1, b2, and b3, and provide more detailed discussions regarding the EBSD images and pole figures.”

Response: Thanks to the reviewers for the valuable reminder.

1. The full name of EBSD, which first appeared, has been provided in the manuscript and highlighted in red.

“Subsequently, the electron back scatter diffraction (EBSD) technology was used to further evaluate the quality of texture (Figs. 2c and d).” **Page 8, Line 17**

2. The captions of Figures 2b₁, b₂ and b₃, which have been changed to c₁, c₂, and c₃, have been provided in the manuscript and highlighted in red. **Page 6**

3. A more detailed discussion on EBSD images and pole figures has been provided in the

manuscript and highlighted in red.

“From the inverse pole figure (IPF) maps and inverse pole/pole figure (IPF/PF) set, it can be seen that the orientation distribution of polycrystals for the T-3.5BHT deviates from random distribution and shows some regularity. For example, almost all grains’ $\langle 00l \rangle_c$ directions are parallel to the z-axis of the sample (perpendicular to the casting direction) instead of x and y-axis (Fig. 2c), and the M_{\max} values in the both IPF/PF set are also greater than 25 (Fig. 2d), indicating a strong $\langle 00l \rangle_c$ texture degree for the T-3.5BHT.” **Page 8, Line 18-25**

Detailed comment 2: “Consider including the data point for the 3.5BHT samples in the composition-temperature phase diagrams since these textured 3.5BHT ceramics exhibit the highest piezoelectric properties.”

Response: Thanks to the reviewers for the valuable reminder. The data points of the T-3.5BHT sample have been added into the composition temperature phase diagram. **(Figs. 2b₁ and b₂)**

Figure 2. The textured mesentropic x BHT ceramics. XRD patterns of the random a₁) and textured a₂) x BHT ceramics. Composition-temperature phase diagrams of the random b₁) and textured b₂) x BHT ceramics. Inverse pole figure maps of $\langle 00l \rangle_c$ textured T-3.5BHT ceramics along c₁) z, c₂) y and c₃) x-axis. d₁) Pole and d₂) inverse pole figure set of the T-3.5BHT ceramics. e₁) Atomic-resolution scanning transmission electron microscopy high-angle annular dark-field (STEM-HAADF) polarization vector image along [100] zone axis for the unpoled T-3.5BHT, and e₂) superimposed with a map of atom polarization vectors.

Detailed comment 3: “Could the authors specify the geometric dimensions of the samples used to test the planar electromechanical coupling factor? Also, it would be helpful if the authors could indicate the resonance and antiresonance frequencies for all samples in Fig. S10.”

Response: Thanks to the reviewers for the good question and valuable reminder.

1. The geometric dimensions of the samples used to test the planar electromechanical coupling factor are ~ 0.3-0.7 mm in thickness and ~ 12-16 mm in electrode diameter.
2. The resonance (f_r) and antiresonance (f_a) frequencies have been provided in Fig. S10, which has been changed to Fig. S12.

Figure S12. Impedance Z and phase angle θ against frequency of the x BHT ceramics measured at room temperature, where f_r and f_a are represented as resonance and antiresonance frequencies, respectively.

Detailed comment 4: “Regarding the comparison of polarization and strain loops, in-situ XRD patterns, SEM images, and PFM images of R-4BHT and T-3.5BHT ceramics, did the authors consider the potential impact of composition on the results?”

Response: We thank the reviewer for the good question. In this work, the addition of an additional 3% NN template resulted in a slight deviation in the composition of the optimal electrical properties of the T- x BHT ceramics from those of R- x BHT, e.g., R-4BHT and T-3.5BHT,

respectively. However, with the variation of the x BHT content, the same trend can be found for both R- x BHT and T- x BHT ceramics in terms of phase structure and a series of electrical property changes, including XRD, Raman, d_{33} , k_p , T_c , $P-E$ and strain loops. Thus, in general, the optimization of the electrical properties of R-4BHT mainly comes from the construction of mesentropy-induced phase boundary, whereas the origin of the high performance of T-3.5BHT ceramics comes from the texture-induced highly $\langle 00l \rangle_c$ orientation in addition to the construction of mesentropy-induced phase boundary. Based on the above analysis we believe that although there are slight compositional differences between R-4BHT and T-3.5BHT, it can be ignored from the point of view of the electrical property enhancement caused by phase boundaries as well as crystal orientation. Therefore, our discussion mainly focuses on the differences in the crystal preferred orientation or random orientation based on constructing the same phase boundary, i.e., NPB, and ultimately selected R-4BHT and T-3.5BHT ceramics with similar phase structures for a comprehensive comparison and analysis.

Detailed comment 5: “How did the authors mitigate the influence of acid etching and the TEM sample preparation process on the domain structure before and after poling?”

Response: We thank the reviewer for the good question.

1. For the preparation of the sample for TEM: Firstly, both unpoled and poled T-3.5BHT samples need to be glued to the glass by hot melt adhesive for overall thinning to ~ 100 - $150 \mu\text{m}$. The Curie temperature of the T-3.5BHT is around $260 \text{ }^\circ\text{C}$, and in order to minimize the impact on the poled T-3.5BHT sample, the heating temperature of the hot melt adhesive is controlled at around $100 \text{ }^\circ\text{C}$ and as low as possible. Next, ion thinning is required, a process that usually also leads to an increase in the temperature of the sample, which can reach tens of degrees or $100 \text{ }^\circ\text{C}$. To further reduce the impact of temperature on the poled sample, we performed liquid nitrogen assistance. **Supporting Information Page 4**
2. For both unpoled and poled T-3.5BHT with chemical corrosion of domain: Firstly, it is also necessary to fix them with hot melt adhesive during polishing. In order to minimize the impact on the poled T-3.5BHT sample, the heating temperature of the hot melt adhesive is controlled at around $100 \text{ }^\circ\text{C}$ and as low as possible. Then, in the chemical corrosion process in order not to cause excessive or insufficient corrosion of the domains, we conducted tests and observed the acid corrosion of the domains at different operating times, and finally determined that the

optimal time for the corrosion of the domains for both unpoled and poled T-3.5BHT was around 2 minutes 15s-30s. **Supporting Information Page 4**

Detailed comment 6: “The temperature stability of the piezoelectric performance of the T-3.5BHT sample is not mentioned. Could the authors provide insights into this aspect?”

Response: We thank the reviewer for the good question, and the *in-situ* temperature-dependent d_{33} value of the T-3.5BHT is provided in Fig. S11. Since the NPB phase boundary belongs to the temperature-dependent phase boundary (*Chem. Soc. Rev.*, 2020, 49, 671-707), the phase structure of the T-3.5BHT ceramics gradually deviates from the phase boundary with the increase of the temperature. The d_{33} value of the piezoceramics is closely related to the phase structure, so the d_{33} value of T-3.5BHT ceramics decreases firstly with increasing temperature due to the deviation of the phase boundary, then stabilizes after 100 °C, and finally decreases sharply due to the generation of the paraelectric phase near the Curie temperature of 260 °C (Fig. S11). The corresponding explanations and descriptions have been provided in the manuscript and highlighted in red. **Page 10, Line 16-19 and Supporting Information Page 18**

Figure S11. *In situ* temperature-dependent d_{33} value of the T-3.5BHT ceramics.

Detailed comment 7: “Given that the transducer performance is closely tied to the fabrication process and the selection of matching and backing layers, could the authors elaborate on how they considered the selection of materials for the matching layer and backing layer? Additionally, what advantages does the transducer performance obtained in this manuscript have compared to that made by other lead-free piezoelectric materials?”

Response:

Generally speaking, the matching layer of a transducer is designed to improve acoustic emission efficiency and increase bandwidth, while the backing layer is mainly used for sound absorption and shock absorption, which can reduce the drag of the transducer and also play a role in increasing bandwidth. There are many types of transducers, and the materials and functions of the matching layer and backing layer are also different. The knowledge inside is too extensive, so sorry that we only know a little bit about it.

1. For the preparation of transducer devices, the sound velocity and density of the matching layer and backing layer should be simulated through finite element simulation based on the parameters of the piezoelectric elements used (including acoustic impedance, sound velocity, density, dielectricity, piezoelectricity, dimensions, etc.). However, the characteristics of common materials hardly match the theoretical values of the simulated matching layer and backing layer. Therefore, we generally use the preparation of composite materials to prepare matching layers and backing layer, which is beneficial for regulating the matching sound speed and density. Composite materials can adjust their acoustic properties by changing the proportion of each component to meet theoretical requirements. The matching layer and backing composite material matrix generally choose epoxy resin materials with stable performance, high strength and good toughness. There are many materials to choose from for filling particles, including both metallic and non-metallic materials. Different particle materials can be added according to different acoustic characteristics requirements. For example, aluminum oxide particles can be selected for the low acoustic impedance layer, copper powder can be selected for the medium acoustic impedance layer, and tungsten powder can be selected for the high acoustic impedance layer. In addition, when selecting fillers, it is also necessary to consider the influence of the particle size of the filler on the matching layer and backing. For example, a particle size that is too large can scatter sound waves, while a particle size that is too small can cause agglomeration. Finally, in this study, based on the acoustic impedance (25.02 MRayls), sound velocity (5610 m/s), density (4.46 g/cm³), dielectricity, piezoelectricity, dimensions and other parameters tested on the T-3.5BHT ceramic, we selected epoxy resin as the composite matrix for both matching layer and backing material, with alumina particles as the filling particles for the matching layer and tungsten powder as the filling particles for the backing material.

2. Due to different transducer structures, different performance can be exhibited. Therefore, the simplicity of the transducer shown in this work cannot be directly compared with other types of lead-free piezoelectric ceramic transducers with different structures. From the perspective of original electrical properties, T-3.5BHT ceramics exhibit excellent comprehensive performance compared to the vast majority of other lead-free piezoelectric ceramics (i.e., BCTZ, BNT and KNN based piezoelectric ceramics, Fig. 3d), including high d_{33} , high k_p and high Curie temperature. Thereinto, high d_{33} is conducive to the sensitivity of the transducer, high k_p help a wide bandwidth, and high Curie temperature is beneficial to temperature resistance characteristics.

Detailed comment 8: “Additional minor comments:”

1. The image in Figure S6a appears to be blurry; kindly consider increasing the resolution of the picture for better clarity.

Response: Thanks to the reviewers for the valuable reminder. To avoid the ambiguity of Fig. S6a, we split Fig. S6 into Figs. S6 and S7 in the manuscript, as shown below:

Figure S6. Energy dispersive spectroscopy (EDS) point analysis of microregion based on transmission electron microscope (TEM) of the T-3.5BHT ceramics.

Figure S7. Atomic percentage of microregions based on EDS point analysis via TEM of the T-3.5BHT ceramics.

- Regarding the statement, “When $1 < S_{\text{config}} < 1.5R$, it is defined as medium entropy,” it is suggested to modify it to “When $1R < S_{\text{config}} < 1.5R$,...” for clarity.

Response: Thanks to the reviewers for the valuable reminder. The sentence of “When $1 < S_{\text{config}} < 1.5R$, it is defined as medium entropy,” has been modified into “When $1R < S_{\text{config}} < 1.5R$,...”.

Page 6, Line 8

- In Figure 4h₂, please verify whether the annotation for E is redundant.

Response: Thanks to the reviewers for the valuable reminder. In Figures 4h₂ and i₂, which have been changed to Figures 4b₁ and c₁, the annotation for E has been removed.

Referee #2:

General comment: “In this manuscript, the author comprehensively studied the effects of entropy and crystal orientation on the electrical properties of KNN-based ceramics through ingenious experimental design. Consequently, giant piezoelectric properties with $d_{33} \sim 680 \pm 35$ pC/N and k_p

~ 72.5%, as well as high $T_c \sim 250$ °C, were obtained in the newly developed mesentropic T-3.5BHT ceramics. The results are very exciting, and I would recommend for publications in NC after minor revisions.”

General response:

We appreciate your positive and constructive comments and suggestions. We have studied the comments and tried our best to revise the manuscript according to your carefully and the corrections are listed below point by point:

All changes in the revised manuscript are highlighted in red.

The followings are the responses for the comments by the reviewers for the manuscript (NCOMMS-23-62921):

Detailed comment 1: “With so many additives in the formula, what is the role of each component?”

Response: We thank the reviewer for the good question. The construction of new phase boundary (NPB) requires both T_{R-O} and T_{O-T} to be moved to or near room temperature simultaneously. Thus, choosing the appropriate additives and tailoring their content are the most important factors for constructing the NPB. It was found that Sb^{5+} , $CaZrO_3$ and $Bi_{0.5}A_{0.5}BO_3$ ($A = K/Na$, $B = Hf/Ti$) all contribute to the decrease of T_{O-T} and the increase of T_{R-O} . The difference is that both Sb^{5+} and $CaZrO_3$ are more focused on the increase of T_{R-O} , while $Bi_{0.5}A_{0.5}BO_3$ ($A = K/Na$, $B = Hf/Ti$) is more focused on the increase of T_{O-T} (*Chem. Soc. Rev.*, 2020, 49, 671-707./ *Chem. Rev.* 2015, 115, 2559-2595). Therefore, the involved Sb, Zr, Hf, Ti, Ca and Bi elements are utilized, which takes into account not only the modulation of atomic configuration entropy S_{config} , but also their respective ability to move the phase boundaries. The corresponding explanations and descriptions have been provided in the manuscript and highlighted in red. **Supporting Information Page 8**

Detailed comment 2: “The dielectric constant of the textured ceramics is not higher than that of random ceramic components, and even lower. Can you further explain?”

Response: We thank the reviewer for the good question. The reason of suppressing dielectric constant by textured engineering is due to two dominant factors (*Adv. Funct. Mater.* 2020, 30, 2001846). The first is the elastolectric composite effect, caused by interfacial stresses due to the lattice mismatch between KNN matrix and the NN templates. The second is associated with the discrepancy in electrical properties. Because the introduced NN templates has a lower dielectric

constant than x BHT ceramics, thus suppressing or even lowering the dielectric constant of the T- x BHT ceramics. The corresponding explanations and descriptions have been provided in the manuscript and highlighted in *red*. **Page 11, Line 3-4 and Supporting Information Page 10**

Detailed comment 3: “The schematic in Figure 1 shows that the grains of the textured ceramics grow along the vertical casting direction, can such information be seen in the SEM?”

Response: We thank the reviewer for the good question. From Figs. S4c and f, it can be seen that the grains of the T-3.5BHT ceramics grow along the vertical casting direction.

Figure S4. The surface (a) and cross-sectional (b) morphology of the R-4BHT ceramics. (c) The cross-sectional morphology of the T-3.5BHT ceramics calcined at 850 °C. (d) The SEM image of the NN templates. The surface (e) and cross-sectional (f) morphology of the T-3.5BHT ceramics calcined at 845 °C.

Detailed comment 4: “TEM is used observe the evolution of the domain structure before and after polarization. Please describe the details. How do you ensure that the polarization state is not damaged during the sample preparation process and the characterization stage (e.g., by the electron beam).”

Response: We thank the reviewer for the good question. For the preparation of the sample for TEM: Firstly, both unpoled and poled T-3.5BHT samples need to be glued to the glass by hot melt adhesive for overall thinning to ~ 100-150 μm. The Curie temperature of the T-3.5BHT is around

260 °C, and in order to minimize the impact on the poled T-3.5BHT sample, the heating temperature of the hot melt adhesive is controlled at around 100 °C and as low as possible. Next, ion thinning is required, a process that usually also leads to an increase in the temperature of the sample, which can reach tens of degrees or 100 °C. To further reduce the impact of temperature on the poled sample, we performed liquid nitrogen assistance.

Detailed comment 5: “The stability of the piezoelectric properties over time is crucial. Can you do a cycling test for the piezoelectric properties?”

Response: We thank the reviewer for the good question. To test the stability of piezoelectric properties over time, we tested the T-3.5BHT ceramic poled one year ago and found no significant depolarization. Furthermore, we also investigated the *in-situ* temperature-dependent d_{33} value of the T-3.5BHT. As shown in Fig. S11, due to the TC of T-3.5BHT being ~ 250 °C, its d_{33} can be well maintained above ~ 600 pC/N before 200 °C, which is beneficial for practical applications. The corresponding explanations and descriptions have been provided in the manuscript and highlighted in red. **Page 10, Line 16-19** and **Supporting Information Page 18**

Figure S11. *In situ* temperature-dependent d_{33} value of the T-3.5BHT.

Referee #3:

General comment: “The development of high-performance lead-free KNN-based piezoceramics is very important for achieving environmentally sustainable society. This work employed a medium entropy strategy to design NPB and utilize texture engineering to induce crystal

orientation. The newly developed KNN-based ceramics enjoys both prominent piezoelectricity and mechanical coupling factor ($d_{33} \sim 680 \pm 35$ pC/N, $k_p \sim 72.5\%$), thus exhibiting an ultrahigh energy harvesting performance ($W_{\text{out}} \sim 4.00$ mW, $P_D \sim 57.90$ $\mu\text{W}/\text{mm}^3$) as well as excellent transducer performance. This work is interesting and the results are very good. The author also gives a deep insight explanation for the observed results. However, a mandatory revision is required. Here are some comments.”

General response:

We appreciate your positive and constructive comments and suggestions. We have studied the comments and tried our best to revise the manuscript according to your carefully and the corrections are listed below point by point:

All changes in the revised manuscript are highlighted in red.

The followings are the responses for the comments by the reviewers for the manuscript (NCOMMS-23-62921):

Detailed comment 1: “One of the main topic of this work is employing a medium entropy strategy to design NPB. It is still unclear what is the advantage of this strategy when comparing other methods of designing NPB? Because the d_{33} is even much lower than some reported KNN based materials with composition mediated NPB with d_{33} over 550pC/N. Furthermore, what is the relationship between it and the following texture process.”

Response: We thank the reviewer for the good question.

1. Recently, more and more research has focused on high-performance high-entropy lead-free ferroelectric energy storage ceramics, but there is a lack of relevant discussion in lead-free piezoelectric ceramics. The high-entropy strategy has emerged as an effective and flexible approach for boosting physical properties (e.g., ferroelectric energy storage) in high-entropy ferroelectrics via the delicate design of local polarization configurations and other intrinsic effects caused by entropy increment, such as entropy stabilization, lattice disorder, inhibition of grain coarsening, improved mechanical properties, cocktail effect, and so on (*InfoMat.* 2023;5:e12488.). However, although the local polymorphic distortion including R-O-T-C multiphase nanoclusters coexistence induced by high entropy strategy in ferroelectric ceramics can greatly reduce the free energy and promote polarization switching, it cannot maintain polarization after removing the electric field due to excessive relaxor, resulting in a contradiction with piezoelectricity (*Nat. Commun.* 13, 3089, 2022.). Therefore, we attempt to

design the new phase boundary (NPB) with only R-O-T related local polymorphic distortion through medium entropy to promote polarization switching while maintaining appropriate residual polarization. Our experimental results also finally prove that the expected NPB can be achieved with the medium entropy strategy, and the piezoelectric properties are optimized. As for the issue of the d_{33} value obtained solely by the medium entropy strategy being lower than other reported KNN based materials with composition mediated NPB, we believe that further optimization can be achieved in the future as this strategy is still in the exploratory stage.

2. Our innovation lies not only in attempting to design NPB using the medium entropy strategy, but also in addressing the inherent drawbacks of KNN-based random ceramics with NPB through subsequent crystal orientation. Therefore, NPB and texture process are a pair of synergistic optimization relations for the comprehensive piezoelectricity of KNN-based ceramics. We mentioned in the Introduction that: “However, the construction of phase boundaries also brings some new issues. On the one hand, achieving higher piezoelectricity inevitably requires sacrificing the Curie temperature. Furthermore, the presence of short-range polar nanoregions (PNRs) within the NPB is not entirely favorable to the piezoelectricity of KNN-based ceramics. Although PNRs facilitate domain switching, it is difficult to achieve saturated poling due to the large energy difference between PNRs and the ferroelectric matrix, and the high content of “non collinear” PNRs, thus, a satisfactory mechanical coupling factor ($k_p < 0.6$) cannot be achieved.”. Therefore, ultimately benefiting from the synergistic optimization of the formation of NPB and coarse grains with $\langle 00l \rangle_c$ orientation, excellent comprehensive piezoelectric performance ($d_{33} \sim 680 \pm 35$ pC/N, $k_p \sim 72.5\%$, $T_c \sim 250$ °C) was obtained in the newly developed mesentropic T-3.5BHT ceramics.

Detailed comment 2: “The order of some figures are confusing. (1) The order of the figures in Figure 2 should be rearranged. According to the reading habit, the figure 2a is better on the left of Figure 2. (2) The Figures 4a and b for STEM appear before Figure 3, which makes the readers very confusing and eyes jump in very side of the manuscript during reading. The figures are suggested to rearranged to make the reader read easier and more comfortable.”

Response: Thanks to the reviewers for the valuable reminder. To make reading easier and more comfortable for readers, we have rearranged Figs. 2 and 4 in the manuscript. As shown below:

Figure 2. The textured mesentropic x BHT ceramics. XRD patterns of the random a_1) and textured a_2) x BHT ceramics. Composition-temperature phase diagrams of the random b_1) and textured b_2) x BHT ceramics. Inverse pole figure maps of $\langle 001 \rangle_c$ textured T-3.5BHT ceramics along c_1) z, c_2) y and c_3) x-axis. d_1) Pole and d_2) inverse pole figure set of the T-3.5BHT ceramics. e_1) Atomic-resolution scanning transmission electron microscopy high-angle annular dark-field (STEM-HAADF) polarization vector image along $[100]$ zone axis for the unpoled T-3.5BHT, and e_2) superimposed with a map of atom polarization vectors.

Figure 4. Electric field-induced multiscale polarization configurations transformation. a) *In-situ* electric field XRD for T-3.5BHT ceramics. b) Transmission electron microscopy (TEM) bright-field images and c) SEM images of acid-etched domain for unpoled and poled T-3.5BHT ceramics. d₁) STEM-HAADF and ABF images along [110] zone axis for the poled T-3.5BHT, and d₂) superimposed with a map of atom polarization vectors. e) Schematic of the interdigital electrode used for *in-situ* electric field piezoresponse force microscopy (PFM). f) *In-situ* electric field optimized vertical PFM (OV-PFM) images for T-3.5BHT ceramics.

Detailed comment 3: “Figures 4c and d are not described in the main text.”

Response: Thanks to the reviewers for the valuable reminder. Figs. 4c and d have been changed to Figs. 4d₁ and d₂. The relevant descriptions are as follows:

“Note here that we also observe evidence of a transformation from the O/T-phase polarization vectors to R phase in the poled T-3.5BHT at the atomic level. This is because there is a microregion structure dominated by R-phase polarization vectors in the poled T-3.5BHT, with only trace amounts of O/T-phase polarization vectors are embedded in the R-phase matrix (Figs. 4d₁ and d₂), which contrasts with the initial one that have a higher proportion of T-phase polarization vectors (Fig. 2e), thus further validating the electric-field-induced phase transition behavior of T-3.5BHT.” **Page 14, Line 18-24**

Detailed comment 4: “The chemical form may be wrong. If the x is larger than 1, the amount of some elements are minus, please check them carefully. Furthermore, from the chemical form, the change of the elements is very small, it may very hard to precisely to weight them. As we know, the phase boundary of KNN based material are very sensitive to the composition, how can you keep the consistence of the performance.”

Response: Thanks to the reviewers for the valuable reminder.

1. The % after x for the chemical form has been supplemented in the manuscript. **Page 6, Line 4**
2. Firstly, in order to maintain accurate weighing and experimental consistency, raw materials with less content involved are of high purity, such as CaCO₃ (Sinopharm, 99.99%), Bi₂O₃ (Alfa Aesar, 99.975%), HfO₂ (Aladdin, 99.99%), ZrO₂ (Aladdin, 99.99%), TiO₂ (Sinopharm, 99.8%), and Sb₂O₃ (Alfa Aesar, 99.9%). Secondly, in order to compensate for the high temperature volatilization of K, we also carried out an additional 1% excess. Last but not least, the two-step sintering method we adopted is also a key tool to effectively improve the reproducibility of the preparation of KNN-based ceramics. **Supporting Information Page 2**

Detailed comment 5: “In page 10, the author said “Both the positive/negative electrostrain of T-3.5BHT reached nearly 0.2% under 30 kV/cm, while that of R-4BHT were only ~ 0.15% and ~ -0.7%, respectively, indicating extremely high piezoelectric activity of T-3.5BHT”. From Fig. S9b, no negative strain with value over -0.2% is observed, how can the large value -0.7% can be obtained? Is it -0.07%? Please double check the data. Furthermore, the value of negative strain should also be given in this sentence for the composition T-3.5BHT.”

Response: Thanks to the reviewers for the valuable reminder. The Fig. 9 has been changed to Fig. 10 and the corresponding error has been fixed as shown below:

“The positive and negative electrostrain of T-3.5BHT can reach nearly ~ 0.2% and ~ -0.19% under 30 kV/cm, respectively, while that of R-4BHT are only ~ 0.15% and ~ -0.07%, indicating extremely high piezoelectric activity of T-3.5BHT (Fig. S10b).” **Page 10, Line 10-12**

Detailed comment 6: “The caption of the figures in Supporting information should be more detailedly described, similar to those of the figures in the main manuscript. Otherwise the reader will be very hard to tell out of the figures, which will affect the wide spread of this work.”

Response: Thanks to the reviewers for the valuable reminder. To make reading easier and more comfortable for readers, the captions of the corresponding Figures have been described in more detailed in supporting information and revised in *red*. **Supporting Information**

Detailed comment 7: “How can the texture engineering improve the k_p ? Please give more discussions.”

Response: We thank the reviewer for the good question. From the experimental results, it can be seen that texture engineering effectively improves both d_{33} and k_p of the x BHT ceramics on the basis of NPB. The improvement mechanism between d_{33} and k_p brought about by texture engineering is closely related, both of which belong to the category of piezoelectric performance improvement and originate from highly saturated poling and high piezoelectric activity. Therefore, the analysis of the piezoelectric performance improvement of high-performance T-BHT ceramics discussed in the manuscript fully covers d_{33} and k_p . For better understanding, we have made corresponding adjustments to the content of the manuscript and as shown below.

1. “This is because the crystal orientation makes the arrangement of polarity vectors more effective under the applied electric field, *thereby improving both d_{33} and k_p simultaneously.*”

Page 10, Line 22-23

2. “To gain more insight into the underlying mechanism of enhanced piezoelectricity, including d_{33} and k_p , we study the electric field-dependent strain/dielectric curves and the internal contributions of ϵ_r” **Page 11-12**
3. “The piezoelectricity is closely related to the electric field-induced local structural phase transition....., which is well consistent to its high saturated poling state.” **Page 12, Line 15-25**
4. “Notably, further comparison reveals that the textured T-3.5BHT has thinner striped domains with higher domain wall density than R-4BHT....., thus greatly improving the saturated poling and piezoelectric response of the T-3.5BHT.” **Page 14-15**

Detailed comment 8: “There are some writing errors in the manuscript, which should be carefully checked. Here are some examples:”

- (1) In line 2 Page 17, “...while the d_{33} has been significantly improves (especially for T-3.5BHT) ...”, the word “improves” should be “improved”.

Response: Thanks to the reviewers for the valuable reminder. The corresponding error has been revised and marked in red in the manuscript. **Page 17, Line 3**

- (2) In Page 12, “... which is well consistent with to its high saturation polarization state and high piezoelectricity...” the word “with” should be deleted.

Response: Thanks to the reviewers for the valuable reminder. The corresponding error has been revised and marked in red in the manuscript. **Page 12, Line 25**

REVIEWERS' COMMENTS

Reviewer #1 (Remarks to the Author):

I am fully satisfied with the authors' response to my comments. The manuscript has significantly improved, and I recommend its publication in its current form.

Reviewer #2 (Remarks to the Author):

The authors have addressed all my comments, I'm happy to recommend for publications in NC.

Reviewer #3 (Remarks to the Author):

All my questions are well addressed, I have no more comments.